# Functional characterization of *Cinnamate 4-hydroxylase* gene family in soybean (*Glycine max*)

**Praveen Khatri**[1,2], **Ling Chen**[1], **Istvan Rajcan**[3], **Sangeeta Dhaubhadel**[1,2]*

**1** London Research and Development Centre, Agriculture and Agri-Food Canada, London, Ontario, Canada,
**2** Department of Biology, University of Western Ontario, London, Ontario, Canada, **3** Department of Plant Agriculture, University of Guelph, Guelph, Ontario, Canada

* sangeeta.dhaubhadel@agr.gc.ca

**Data Availability Statement:** All relevant data are within the paper and its Supporting information files.

## Abstract

Cinnamate 4-hydroxylase (C4H) is the first key cytochrome P450 monooxygenase (P450) enzyme in the phenylpropanoid pathway. It belongs to the CYP73 family of P450 superfamily, and catalyzes the conversion of *trans*-cinnamic acid to *p*-coumaric acid. Since *p*-coumaric acid serves as the precursor for the synthesis of a wide variety of metabolites involved in plant development and stress resistance, alteration in the expression of soybean *C4H* genes is expected to affect the downstream metabolite levels, and its ability to respond to stress. In this study, we identified four *C4H* genes in the soybean genome that are distributed into both class I and class II CYP73 family. *GmC4H2*, *GmC4H14* and *GmC4H20* displayed tissue- and developmental stage-specific gene expression patterns with their transcript accumulation at the highest level in root tissues. *GmC4H10* appears to be a pseudogene as its transcript was not detected in any soybean tissues. Furthermore, protein homology modelling revealed substrate docking only for GmC4H2, GmC4H14 and GmC4H20. To demonstrate the function of GmC4Hs, we modified a cloning vector for the heterologous expression of P450s in yeast, and used it for microsomal protein production and enzyme assay. Our results confirmed that GmC4H2, GmC4H14 and GmC4H20 contain the ability to hydroxylate *trans*-cinnamic acid with varying efficiencies.

## Introduction

Plant cytochrome P450 monooxygenases (P450) are heme thiolate proteins belonging to one of the largest and functionally diverse superfamilies of enzymes. Even though the majority of the P450s perform oxidation reactions, some P450s catalyze a variety of other reactions such as sulphooxidations, dealkylations, epoxidations, peroxidations, isomerizations and cyclizations [1, 2]. Plant P450s play important roles in a wide spectrum of metabolic processes leading to the production of both primary and specialized metabolites including phenylpropanoids. The phenylpropanoid biosynthetic pathway involves multiple hydroxylation steps and utilizes P450s belonging to multiple different families. Cinnamate 4-hydroxylase (C4H), a member of

**Funding:** This research was supported by the Agriculture and Agri-Food Canada's Genomics Research and Development Initiative grant (J-002364) and the ASC-09 Soybean Cluster Activity #7A (J-002060) to SD.

**Competing interests:** The authors have declared that no competing interests exist.

CYP73 family, is the first key P450 enzyme in the phenylpropanoid biosynthetic pathway that catalyzes the conversion of *trans*-cinnamic acid to *p*-coumaric acid (Fig 1A). The early phenylpropanoid pathway involves three enzymes: (i) Phenylalanine ammonia lyase (PAL) catalyzes the deamination reaction converting L-phenylalanine to *trans*-cinnamic acid; (ii) C4H uses *trans*-cinnamic acid as a substrate and produces *p*-coumaric acid; (iii) *p*-coumaric acid is used as a substrate by 4-coumarate coenzyme A ligase (4CL) converting it into *p*-coumaroyl-CoA. This is also known as general phenylpropanoid pathway. *p*-Coumaroyl-CoA is a major branch point metabolite that provides the precursor for channeling carbon flow for the synthesis of a wide variety of metabolites such as (iso)flavonoids, monolignols, lignans and sinapate esters (Fig 1B) [3, 4].

Genes encoding C4Hs have been identified and their ability to convert cinnamic acid to *p*-coumaric acid confirmed in a wide variety of plants, including *Arabidopsis thaliana* [5], *Helianthus tuberosus* [6], *Cicer arietinum* [7], *Populus trichocarpa* [8]. Phenylpropanoids include pre-formed and inducible anti-microbial compounds, signal molecules, as well as structural polymers which play important function in plant growth and stress resistance. The importance of these specialized metabolites together with many other plant natural compounds has encouraged the efforts to produce them in heterologous systems by transferring the entire metabolic pathway [9–13]. A P450 enzyme generally requires NADPH: cytochrome P450 reductase (CPR) as a redox partner for its catalytic activities [14, 15]. Both P450s and the CPR are membrane bound proteins and are located in the endoplasmic reticulum (ER) [16, 17]. The membrane localization of eukaryotic P450s makes it challenging to produce them directly in prokaryotic systems [18, 19].

Genes encoding key enzymes in the phenylpropanoid biosynthetic pathway are coordinately activated as a result of stress response such as wounding, exposure to ultraviolet (UV) radiation and pathogen infection. For example, UV irradiation induces the expression of *PAL* and *C4H* and increases their activity during the storage of tomato fruit (*Lycopersicon esculentum*) leading to the increased levels of phenolic compounds [20]. *C4H* expression is induced by cold and salt stress in kenaf (*Hibiscus cannabinus* L.) [21], and drought stress in *Capsicum annum* [22]. Several studies have also reported the induced expression of *C4H* upon pathogen infection in host plants [23, 24].

Soybean (*Glycine max* L. Merr) is an important grain legume contributing 61% of world's total oilseed production in 2022 (http://soystats.com/international-world-oilseed-production/). Soybean seeds contain approximately 40% protein and 20% oil by dry weight [25]. Enhancement of soybean yield and seed protein content without compromising the oil have been the major goals of soybean breeding. However, soybean yield is affected by several environmental factors such as biotic and abiotic stresses. Increased production of several specialized metabolites such as flavonoids, isoflavonoids, terpenoids, saponins, lignins have been found in soybean as a result of stress [26–30] suggesting their roles in stress resistance.

In this study, we performed a genome-wide search of soybean *C4H* (*GmC4H*) and identified four *C4H* genes in the soybean genome. Analysis of GmC4Hs for their predicted protein structure, the presence of conserved P450 motifs and tissue-specific gene expression profiles suggested three out of four GmC4H were functional. Using yeast heterologous expression system for recombinant GmC4H production and enzyme assays, we confirm that three GmC4H enzymes catalyze the conversion of *trans*-cinnamic acid to *p*-coumaric acid but differ in their catalytic efficiencies. Our findings provide further insight into the function of GmC4Hs in plant development and during stress.

**A**

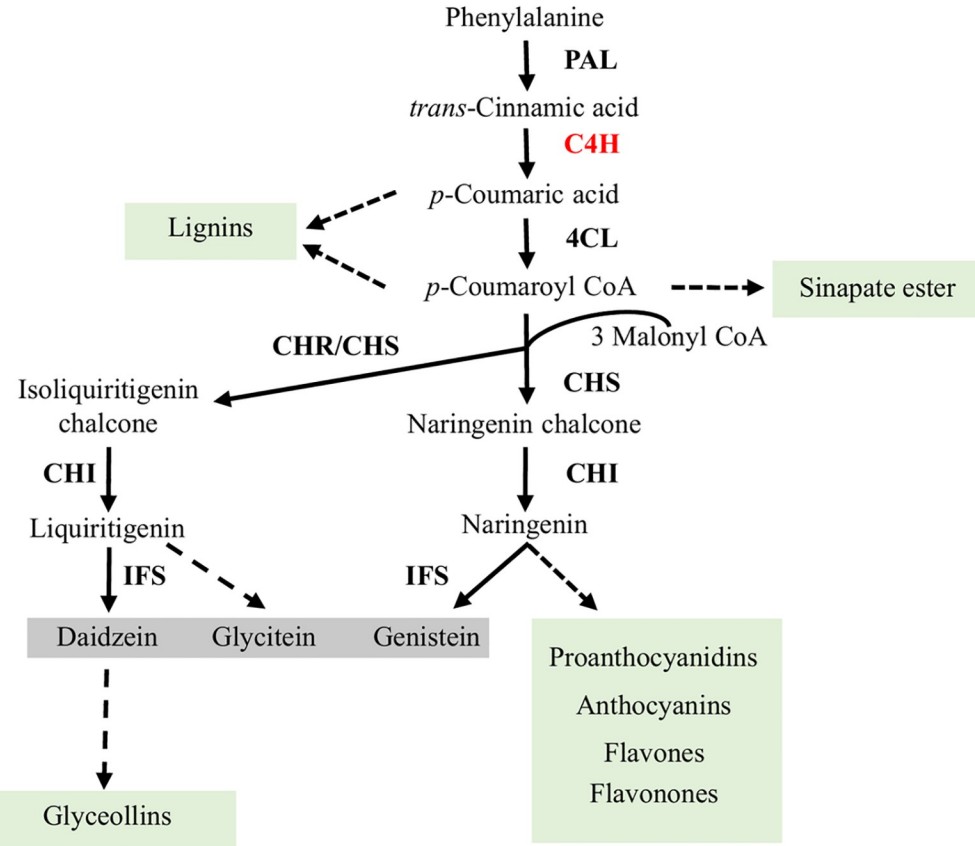

**B**

**Fig 1. Phenylpropanoid biosynthetic pathway.** (A) C4H catalyzes the conversion of trans-cinnamic acid to *p*-coumaric acid by adding the hydroxyl group at the 4′ position. (B) A schematic diagram depicting soybean phenylpropanoid pathway highlighting the involment of C4H (red) leading to biosynthesis of specialized metabolites such as isoflavones (daidzein, genistein, glycitein) and other end products (green highlights).

## Materials and methods

### Plant materials and chemicals

Soybean (*Glycine max* L. Merr) cultivar Harosoy63 seeds were used from collections at Agriculture and Agri-Food Canada, and was grown in London research and development center's field plots. The seeds were planted in two consecutive years (2011 and 2012) to collect two biological replicates of seed samples. Pods were randomly selected from five to seven plants, and

developing embryos at 30, 40, 50, 60, and 70 days after pollination (DAP) were excised from the seeds, frozen in liquid nitrogen, and stored at -80˚C.

*Nicotiana benthamiana* seeds were obtained from Dr. Rima Menassa (AAFC-London) and plants were grown in a growth room under a 16 h light at 25˚C and 8 h dark at 20˚C cycle with light intensity of 80–100 μmol m$^{-2}$ sec$^{-1}$ and relative humidity of 60–70%. All local, national or international guidelines and legislation were adhered for the use of plants in this study.

The standards of *trans*-cinnamic acid and *p*-coumaric acid (HPLC grade) were obtained from Sigma-Aldrich. For HPLC analysis, analytical-grade chemicals were obtained from Sigma-Aldrich and Thermo-Scientific.

## Identification of *GmC4H*s and *in silico* analysis

To search for C4H encoding genes in the soybean genome, a BLASTp search was performed using the amino acid sequence of *Arabidopsis thaliana* C4H (TAIR: AT2G30490) as query. Additionally, amino acid sequences of CYP73 members from other plant species were searched in UniProt (https://www.uniprot.org/). Multiple sequence alignment was performed using Clustal Omega [31] and visualized using pyBoxshade alignment tool (https://github.com/mdbaron42/pyBoxshade). Phylogenetic tree was constructed using the maximum likelihood method with a 1000 bootstrap value in Mega 10 [32]. Gene structures were analyzed using GSDS (Gene Structure Display Server 2.0, http://gsds.gao-lab.org/). Protein molecular mass and *p*I were calculated using ExPASy (https://web.expasy.org/compute_pi/) and protein motifs and spacing between motifs were analyzed as described previously [33].

## Homology modeling and molecular docking

For docking study, an AlphaFold2 based plant cytochrome P450 protein structure prediction database (PCPD) was used along with imbedded heme cofactor in the structure [https://p450.biodesign.ac.cn [34]]. To prepare the protein for docking, MGLtools 1.5.7 (https://ccsb.scripps.edu/mgltools/) was used. All the water molecules were deleted from protein structure, polar hydrogens and Kollman charges were added to structure. To prepare the ligand molecule, *trans*-cinnamic acid structure was downloaded from ZINC15 database (ZINC16051516) [35], saved as MOL2 format and converted to pdb format using pymol (https://pymol.org/2/). The gridbox was built near heme moiety by MGLtool included 40 points in each x, y and z directions, with a grid spacing of 0.37 Å. The docking of GmC4Hs and ligand was performed using Autodock vina [36] using energy range 4 and exhaustiveness 8. The docking results for GmC4Hs were superimposed on each other and visualized using BIOVIA Discovery Studio Visualizer (https://discover.3ds.com/discovery-studio-visualizer-download).

## Gene expression analysis and heat map generation

For gene expression analysis, the expression data was collected from the publicly available gene atlas at soybase (https://soybase.org/soyseq/). The candidate gene IDs were used as a search query in the database and normalized reads from root, nodule, young leaf, flower, one cm pod, pod shell (10 and 14 days after fertilization), and seed (10, 14, 21, 25, 28, 35, 42 days after fertilization) tissues were obtained. The heat map was generated using TBtools v1.10056. The normalized reads were clustered hierarchically using a complete linkage method and distance were calculated using the Euclidean method.

## RNA extraction and quantitative RT-PCR analysis

To explore the expression level of *GmC4Hs* in different embryo developmental stages, total RNA was isolated using RNeasy Plant Mini Kit (Qiagen, USA) with on-column DNA digestion with DNase 1 enzyme (Thermo Scientific, USA) and quantified using a NanoDrop spectrophotometer (Thermo Scientific, USA). cDNA was synthesized from 1 μg of total RNA using SuperScript IV Reverse Transcriptase (Invitrogen, USA). Quantitative RT-PCR was performed using a real-time PCR detection system (Bio-Rad, USA) together with SsoFastTM EvaGreen® Supermix (Bio-Rad, USA) and gene-specific primers (S1 Table). The *CONS4* [37] was used as a reference gene for data normalization using CFX Maestro (Bio-Rad, USA).

## Modification of yeast expression vector

To create a Gateway compatible yeast expression vector (pESC-Leu2d-LjCPR1-GW) for P450 expression in yeast, the backbone of a pESC-Leu2d vector was used. The *Lotus japonicus Cytochrome P450 reductase 1* (*LjCPR1*) gene fragment and Gateway cassette were ligated in the MCS1 and MCS2 of pESC-Leu2d, respectively (Fig 2). A 2121 bp coding region of *LjCPR1* gene (Accession no. AB433810) was amplified from cDNA prepared from *L. japonicus* root RNA using gene-specific primers with restriction enzyme sites NotI and PacI incorporated in the forward and reverse primers, respectively (S1 Table), and cloned into the pGEM-T easy vector (Promega corp, USA) to obtain pGEM-LjCPR1. Both pESC-Leu2d and pGEM-LjCPR1 vectors were digested with NotI and PacI restriction enzymes and purified *LjCPR1* fragment was ligated into the MCS1 at the NotI and PacI site of pESC-Leu2d to obtain pESC-Leu2d-LjCPR1. To make the pESC-Leu2d-LjCPR1vector gateway compatible, a Gateway cassette (1725bp) was amplified from the pEarleyGate100 vector [38] using the ApaIattR1 and SacIIattR2 primers (S1 Table) and the resulting PCR product was sub-cloned into the pGEMT vector to obtain pGEMT-GW. pGEMT-GW and pESC-Leu2d-LjCPR1 vectors were digested with ApaI and SacII to insert the Gateway cassette into MCS2 of the pESC-Leu2d-LjCPR1 to obtain pESC-Leu2d-LjCPR1-GW (Fig 2). For all the transformation and plasmid multiplication, *Escherichia coli* DH5α was used.

## Gene cloning

For gene cloning, full-length *GmC4Hs* were amplified from soybean cDNA using gene-specific primers (S1 Table) and Platinum SuperFi II Polymerase (Invitrogen, USA). The PCR products were cloned into the gateway entry vector pDONR-Zeo (Invitrogen, USA) using the BP clonase reaction mix (Invitrogen, USA), transformed into *E. coli* DH5α, and grown on LB media supplemented with zeocin (50 μg/mL). The *E. coli* colonies containing recombinant plasmids were screened by colony PCR using gene-specific primers (S1 Table) to get pDONRZ-GmC4H. The entry clones were sequenced for confirmation followed by recombination into pESC-Leu2d-LjCPR1-GW using LR clonase reaction mix (Invitrogen, USA) for enzyme assay or pEarleyGate101 for subcellular localization. The recombinant plasmids were transformed into *E. coli* DH5α, screened by colony PCR using gene-specific primers. Positive pESC-Leu2d-LjCPR1-GW-GmC4H plasmid DNA was transformed into *Saccharomyces cerevisiae* strain BY4742 using Frozen-EZ Yeast Transformation II Kit (Zymo Research, USA). Positive pEarleyGate101-GmC4H plasmid DNA was transformed into *Agrobacterium tumefaciens* GV3101 for subcellular localization.

## Subcellular localization and confocal microscopy

Subcellular localization of GmC4H20 was studied by infiltrating *A. tumefaciens* GV3101 carrying pEarleyGateG101-GmC4H20 into *N. benthamiana* leaves, as described previously [39].

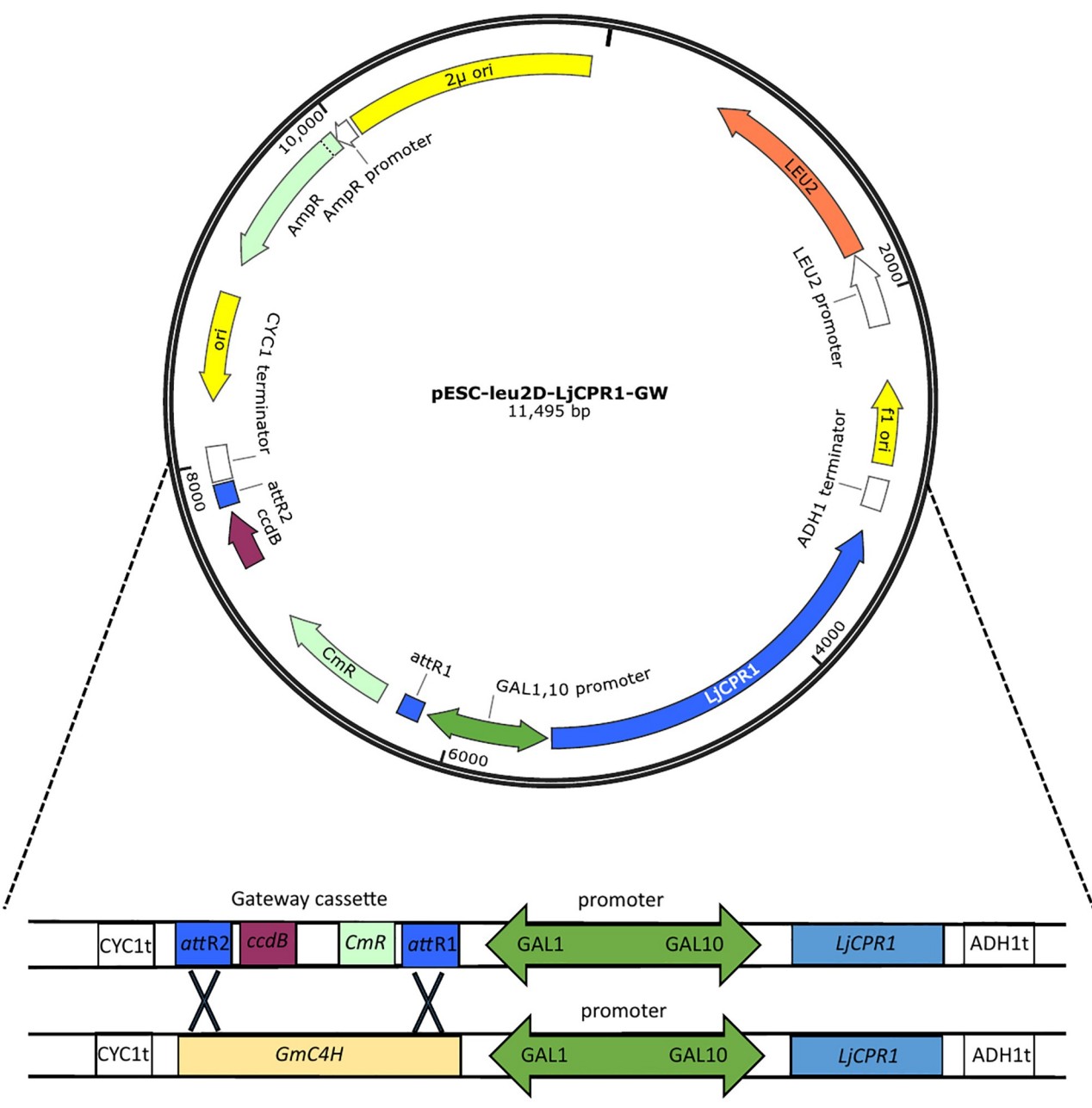

**Fig 2. A map of pESC-leu2D-LjCPR1-GW yeast expression vector for P450 assay.** The bidirectional Gal1/Gal10 promoter of pESC-Leu2d vector drives *LjCPR1* and gene cloned within the gateway cassette.

Transient expression of protein was visualized by confocal microscopy using a Leica TCS SP2 inverted confocal microscope (Leica) at an excitation wavelength of 514 nm and emissions were collected at 530–560 nm with a 63× water-immersion objective for YFP imaging.

## Yeast microsome preparation and enzyme assay

The enzyme assay was performed using microsomes containing recombinant GmC4H protein. Protein induction and microsome preparation were conducted as described previously

[40, 41] with some modifications. *S. cereviceae* containing the recombinant pESC-Leu2d-LjCPR1-GW-GmC4H was grown on minimal synthetic dextrose (SD)/-Leu liquid media (5 mL) for 2 days at 30˚C and 130 rpm. The seed culture was transferred in a 400 mL growth media containing 0.67% yeast nitrogen base, 2% glucose (Sigma-Aldrich, USA) and 0.069% Leucine-dropout supplement (Takara Bio Inc, USA) and grown in a shaker incubator at 30˚C and 130 rpm for 48 hours. After 48 hours, the cell pellet was collected by centrifugation at 7000 rpm at 20˚C. The pellet was washed twice with autoclaved distilled water, suspended in induction media containing 0.67% yeast nitrogen base, 2% galactose, 1% raffinose and 0.069% Leucine-dropout supplement and grown for 24 h at 30˚C and 130 rpm. The cell pellet was re-suspended in 20 mL 100 mM phosphate buffer (pH 7.6) containing 1% protease inhibitor. Cells were lysed using a French Laboratory Press set to 1200 psi. Cell lysates were then centrifuged at $10,000 \times g$ to remove the debris. The supernatant was subjected to ultracentrifugation at $100,000 \times g$ for 1 hour. Microsomes were re-suspended in phosphate buffer (pH 7.6), and microsomal protein concentration was quantitated by Bradford assay [42].

To examine the *in vitro* biochemical properties of GmC4Hs, microsomal proteins (1 mg) were incubated with 50 mM phosphate buffer (pH 7.6) containing 10 μM cinnamic acid and 1 mM NADPH for 15 min at 25˚C. The reaction was stopped with 10 μL of concentrated HCl followed by ethyl acetate extraction and HPLC analysis.

To determine apparent Km values, GmC4H microsomal protein (2–100 μg) was incubated with substrate concentrations ranging from 1–140 μM. Other reaction components were kept same as described above, followed by HPLC. Kinetic constants were calculated from initial rate data using Michalis Menten equation in graphpad prism 8 using nonlinear regression. The reaction product was compared with the authentic standard.

## HPLC analysis

HPLC analysis was performed following methods previously developed using an Agilent 1290 series HPLC [43]. A C18 column was used with a mobile-phase gradient of 25–100% acetonitrile with 0.1% trifluoroacetic acid (TFA) over 8 min at a flow rate of 1 mL min$^{-1}$ and the detector set at 254 nm. Solvent A contained water with 0.1% TFA while solvent B contained acetonitrile with 0.1% TFA. The sample injection volume was 5 μL and column temperature was maintained at 35˚C. The gradient elution was set as follows: 0 min to 3 min, 75% solvent A, 25% solvent B; 3 min to 4.75 min, solvent A was decreased to 15% and solvent B increased to 85%; 4.75 to 5 min, 15% solvent A and 85% solvent B; 5 min to 5.55 min, 100% solvent B until 8 min. The UV spectrum and retention time of the substrate and product were compared to commercial standard *trans*-cinnamic acid and *p*-coumaric acid (Sigma-Aldrich).

## Results

### Identification and sequence analysis of GmC4Hs

To identify GmC4Hs present in soybean, we used the C4H sequence from *A. thaliana* (TAIR: AT2G30490) as a query sequence to perform a BLASTp search against *Glycine max* Wm82.a4.v1 database in Phytozome 13. The search resulted in four hits containing greater than 60% sequence identity with the query sequence. Additionally, a keyword search using the word "cinnamate 4-hydroxylase" also resulted into the same four candidates with gene identifiers *Glyma.02G236500* (*GmC4H2*), *Glyma.10G275550* (*GmC4H10*), *Glyma.14G205200* (*GmC4H14*) and *Glyma.20G114200* (*GmC4H20*). The coding sequences of *GmC4H* family members ranged from 600 bp (*GmC4H10*) to 1617 bp (*GmC4H20*) encoding proteins with molecular masses of 22.3 to 61.3 kDa, respectively. The gene structure analysis revealed similar structure for *GmC4H2* and *GmC4H14* with 4 exons and 3 introns. *GmC4H20* contains one

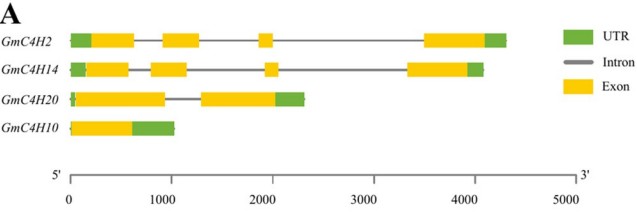

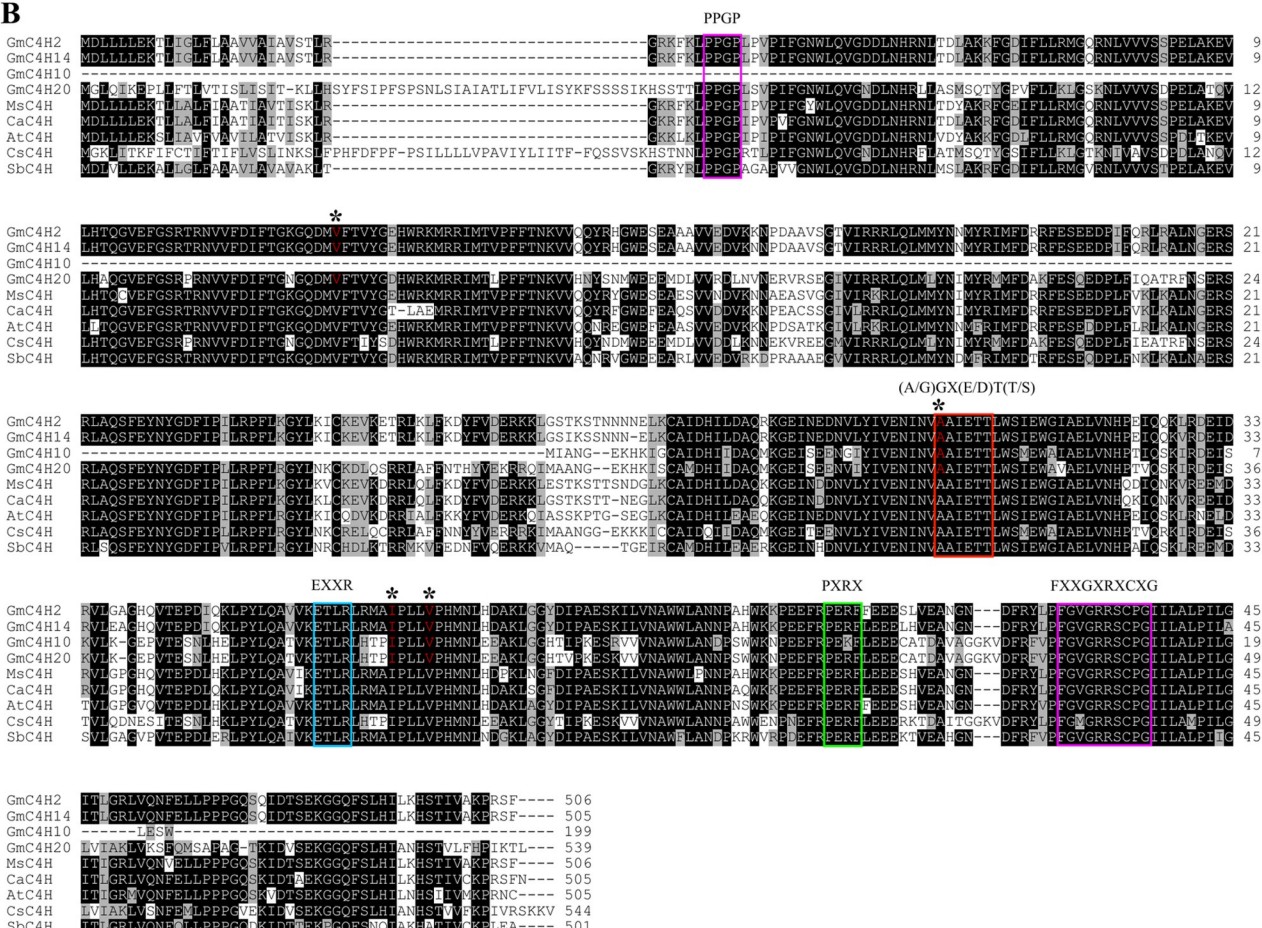

**Fig 3. *GmC4H* gene structures and motif analysis.** (A) Schematic diagram of gene structure of *GmC4Hs*. Exon-intron structures of *GmC4Hs* were compiled from Phyotozome 13 database (https://phytozome-next.jgi.doe.gov/info/Gmax_Wm82_a4_v1) drawn by Gene Structure Display Server 2.0. (B) Multiple sequence alignment of deduced amino acid sequences of candidate GmC4Hs with characterized C4H from other plant species. Identical and similar amino acid residues are indicated by black and gray shades, respectively. Five P450 motifs: the PPGP, the oxygen-binding, the ETLR, the PERF and the heme-binding motifs are indicated by different colored rectangles. Asterisk (*) and red font indicate critical amino acid residues for substrate binding. Accession numbers are as indicated: MsC4H, P37114; CaC4H, O81928; AtC4H, P92994; CsC4H, ASU87408; SbC4H, Q94IP1.

intron and 2 exons while *GmC4H10* is intronless (Fig 3A). The amino acid sequence of GmC4H2 and GmC4H14 shared 98% sequence identity whereas GmC4H20 showed 86.6 and 86.8% sequence identity with GmC4H2 and GmC4H14, respectively. The multiple sequence alignment of GmC4H candidates with functionally characterized C4H from *Arabidopsis thaliana* [5], *Medicago sativa* [24], *Sorghum bicolor* [44], *C. arietinum* [7] and *Camellia sinensis* [45] revealed that all 5 conserved P450 motifs such as proline-rich motif (PPGP) at N terminal,

oxygen binding motif (AAIETT) in I helix and ETLR and PERF motifs in K-helix as well as a heme-binding motif (FGVGRRSCPG) are present in all the candidate GmC4Hs except for GmC4H10 that lacks the PPGP motif. Additionally, GmC4H10 contains an arginine to lysine substitution in PERF motif (Fig 3B).

To investigate the evolutionary relationship between GmC4Hs and CYP73 members from other plant species a phylogenetic tree was constructed. Our search for functionally characterized CYP73 from other plants identified a total of 33 CYP73s (S2 Table). The phylogenetic tree included all the functionally characterized plant CYP73 to date and 174 predicted C4H from a wide range of plant species. As shown in Fig 4, the phylogenetic tree branched into three main groups: CYP73 family members from angiosperms (class I and class II) and CYP73s from gymnosperms, liverworts, mosses, hornworts, and ferns. Interestingly, legume-specific clades were observed in both classes of CYP73 family. GmC4H2 and GmC4H14 cluster together with class I C4H from legumes that contain two functionally characterized CYP73 from *M. sativa* and *C. arietinum* while GmC4H10 and GmC4H20 group together with class II CYP73 members. Among the several predicted C4Hs that belong to the class II, only a single C4H is functionally characterized to date [45].

## Molecular modeling of candidate GmC4Hs

To gain an insight into the structural basis of the candidate GmC4H substrate specificity, we performed homology modeling of all four predicted GmC4Hs using the PCPD database and used AutoDock docking of *trans*-cinnamic acid into the active site cavity of their three-dimensional structures (Fig 5). The results revealed that *trans*-cinnamic acid binds to GmC4H2, GmC4H14, GmC4H10 and GmC4H20 protein molecules with an estimated free binding energy of -6.6, -6.9, -5.8 and -6.7 kcal/mol, respectively. The most predominant orientations for GmC4H2, GmC4H14 and GmC4H20 oriented the alkyl chains towards the heme iron in the active site and showed a Pi-alkyl interaction while GmC4H10 showed no orientation towards the heme-iron (Fig 5, S1 Fig). The hydrophobic interactions were mediated by three amino acid residues Ala307, Val376 and Ile372 in GmC4H2 and Val151, Ala338, Val406 in GmC4H20. GmC4H14 contained four hydrophobic residues (Val118, Ala306, Ile371, and Val375) in the active site interacting with the benzene ring of ligand *trans*-cinnamic acid. The Ala residue of oxygen binding motif of GmC4Hs showed Pi-alkyl interaction with *trans*-cinnamic acid suggesting its participation in substrate binding and conversion. The three dimensional structure of GmC4H10 differs significantly from other candidate GmC4Hs and the heme group was positioned away from *trans*-cinnamic acid docking. The homology modelling of GmC4H raises the possibility that it may not be a functionally active protein as the docking position is impossible for hydroxylation.

## Expression analysis of *GmC4H* gene family members

To evaluate the expression of candidate *GmC4H* genes in soybean tissues, we used the publically available transcriptome data of soybean tissues in Soybase database (https://soybase.org/soyseq/). The dataset consisted of transcript accumulation in multiple soybean tissues such as root, nodule, leaf, flower, pod shell, and seeds at various developmental stages presented in reads per kilobase of transcript per million mapped reads (RPKM) values. As shown in Fig 6A, maximum expression of *GmC4H2*, *GmC4H14* and *GmC4H20* was found in root tissues with RPKM values 255, 63 and 48, respectively. *GmC4H10* transcript was not detected in any of the tissues included in the study. Since the dataset lacks the later seed developmental stages, we performed a qRT-PCR analysis of soybean embryos at different development stages ranging from 30 to 70 days after pollination (DAP). As shown in Fig 6B, *GmC4H*s were expressed in all

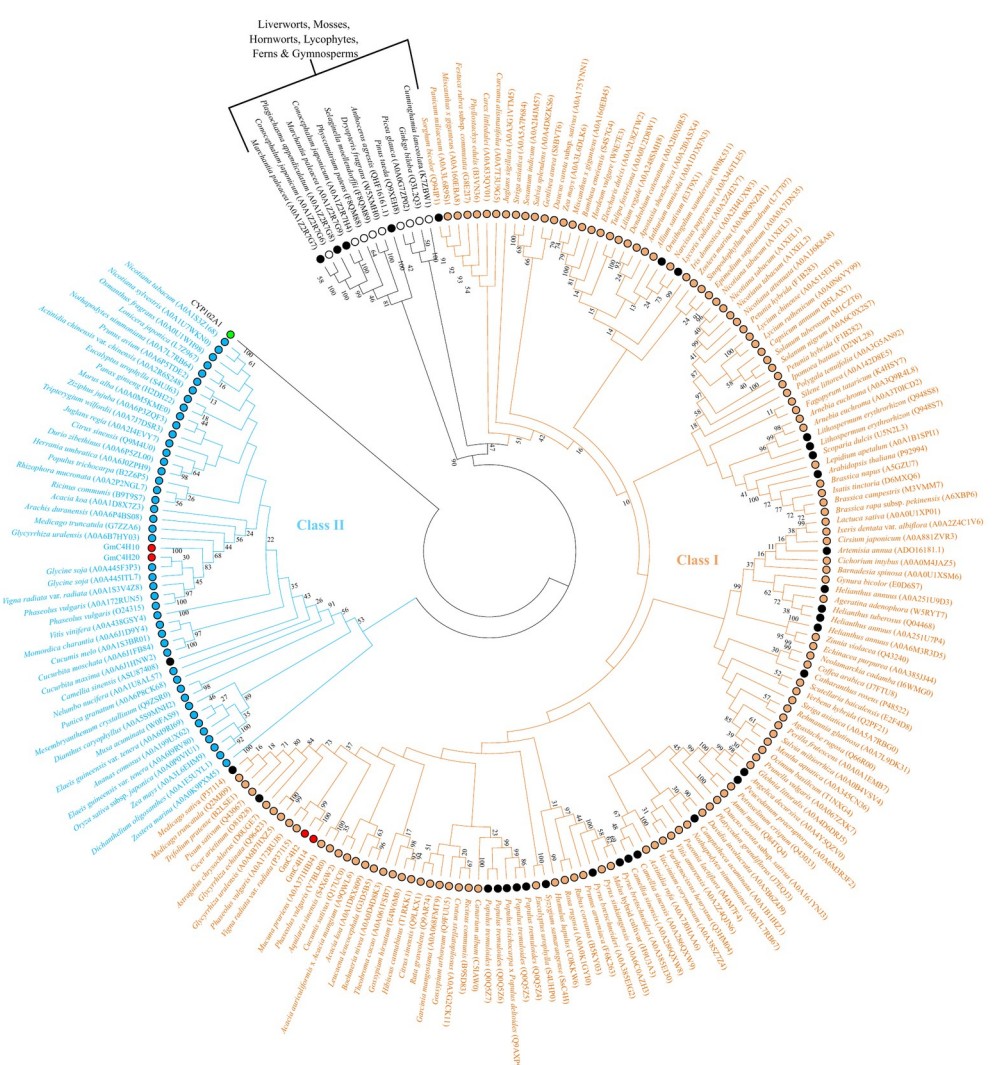

**Fig 4. Phylogenetic analysis of GmC4H proteins.** A maximum likelihood tree was generated by MEGA 10 using the putative amino acid sequences of candidate GmC4Hs and CYP73 candidates from other plant species. Bootstrap replicates of 1000 expressed in percentage are shown next to the branch. Class I and class II CYP73 members are indicated by brown and blue branches of tree, respictively while CYP73 from gymnosperms, mosses, ferns etc are indicated by black branches. Functionally charactersized C4Hs are indicated by black circles and GmC4Hs are shown by red circles. CYP102A1 from *Bacillus megaterium* (accession no. P14779) was used as an outgroup.

embryo development stages. Notably, the expression levels of *GmC4H2* and *GmC4H14* increased from mid (30, 40, and 50 DAP) to late embryo development stages (60 and 70 DAP) while *GmC4H20* transcript accumulation level increased gradually in early stages of seed development, peaked at 50 DAP and then decreased gradually during the late development stages. A drastic increase in relative transcript abundance of *GmC4H2* and *GmC4H14* at 70 DAP and gradual increase in *GmC4H20* transcript accumulation during early embryo development suggests their roles during stage-specific embryo development. As GmC4H10 lacks PPGP conserved motif, show low substrate binding affinity in the docking experiment and its transcript was undetectable in soybean tissues, it was eliminated for further characterization.

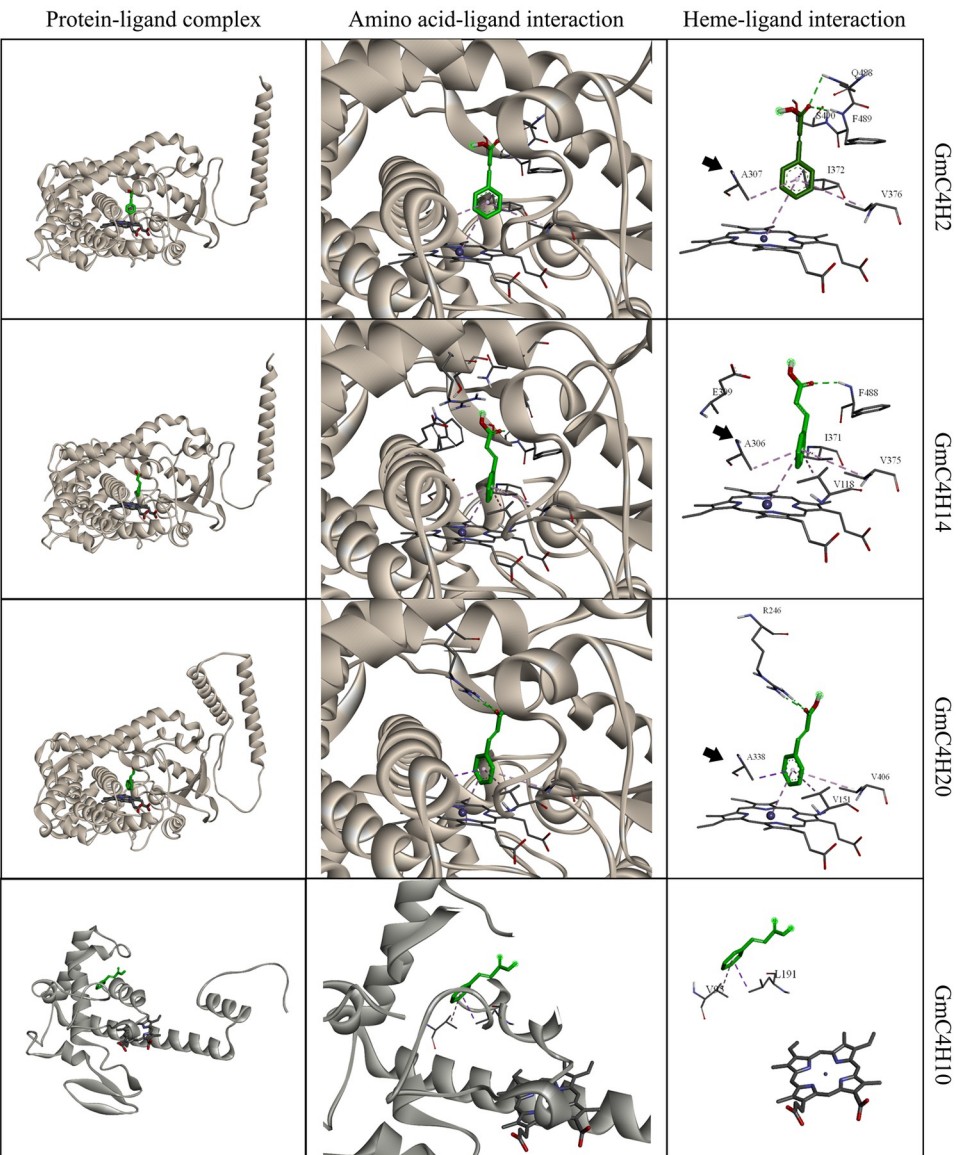

**Fig 5. Molecular docking of GmC4Hs with *trans*-cinnamic acid.** On the Left, predicted 3D structure of GmC4Hs showing position of *trans*-cinnamic acid (green) adjacent to heme-group (blue) in protein-ligand complex. Amino acid residues interacting with ligand are shown with red color in protein structure. Middle column depicts magnified view of ligand interacting with heme-group and amino acid residue in the protein-ligand complex. Interaction of ligand with heme group showing with interacting amino acid residues and heme-group is displayed in heme-ligand interaction (right column). Pink dotted line indicates Pi-Sigma interaction, light pink showing Pi-Alkyl interaction, green showing hydrogen bond and light green showing Van der Waals interactions. Conserved alanine residue in GmC4Hs are indicated by solid black arrow.

## GmC4Hs are localized in the endoplasmic reticulum

Previously, we demonstrated the ER localization of both GmC4H2 and GmC4H14 [46]. Using the DeepLoc sever [47], GmC4H20 was also predicted to localize in the ER. To confirm the subcellular localization of GmC4H20, its coding gene sequence was translationally fused with YFP, and transiently co-expressed with an ER-CFP marker in *N. benthamiana* leaf epidermal

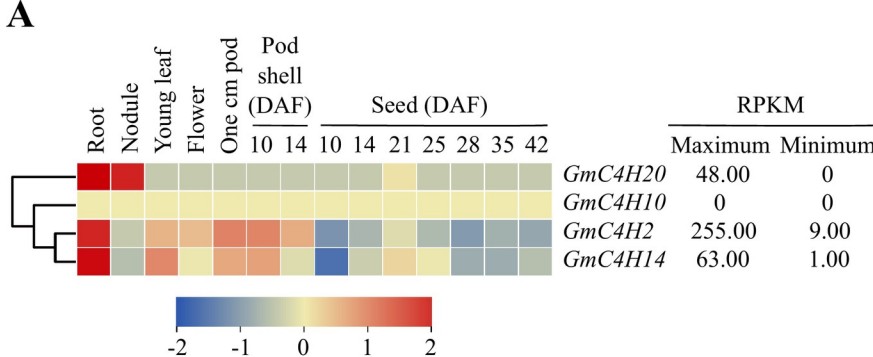

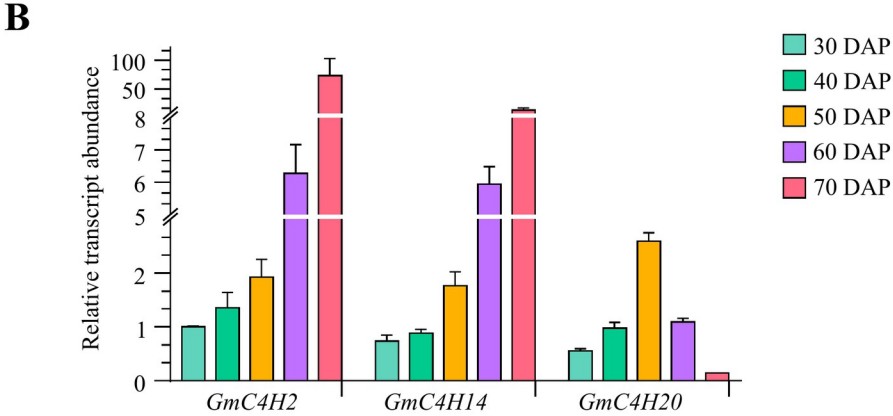

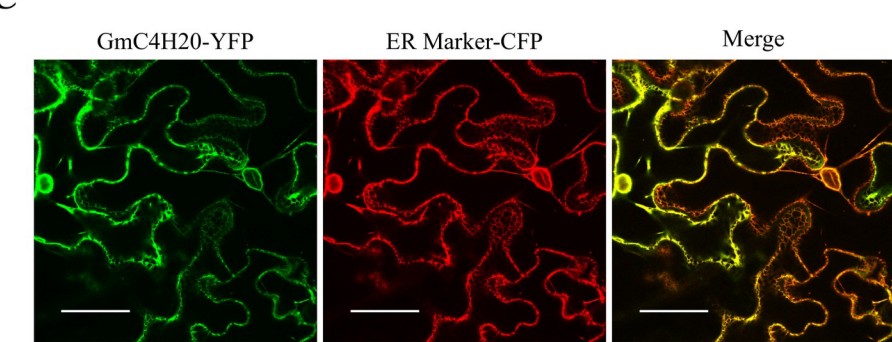

**Fig 6. Expression analysis of *GmC4Hs*.** (A) The GmC4H gene expression data across different tissues and developmental stages were obtained from Soybase (https://soybase.org/soyseq/) for the heatmap generation. The color scale below the heat map indicates expression values, blue and red indicating low and high transcript abundance, respectively. (B) Analysis of *GmC4H* transcript accumulation in soybean embryos during development. cDNA was synthesized from total RNA (1 μg) isolated from soybean embryos (30, 40, 50, 60 and 70 DAP) and qPCR was performed using gene-specific primers. Expression values were normalized against the reference gene *CONS4*. The error bars represent the SEM of two biological replicates and three technical replicates for each biological replicate. (C) Localization of GmC4H20 in subcellular compartment. A translational fusion of GmC4H20-YFP was transiently expressed in the leaf epidermal cells of *N. benthamiana* and visualized by confocal microscopy. Co-localization of GmC4H20-YFP fusion with a ER marker fused to CFP was performed for confirmation. A merged signal was obtained by sequentially scanning the two channels. Scale bar: 50 μm.

cells. As shown in Fig 6C, GmC4H20 was observed in the ER as indicated by a typical reticulate fluorescence pattern of ER localization and the overlapping signal with the ER marker.

## Functional characterization of GmC4Hs

Since GmC4Hs localize in the ER, use of yeast expression system has advantages over prokaryotic systems for their functional characterization. Additionally, in eukaryotes, P450's catalytic activity requires a CPR to provide electron from a redox partner NADPH [15]. To investigate the functional properties of GmC4Hs, we created a gateway compatible vector pESC-Leu2d-LjCPR1-GW containing *LjCPR1* under a bidirectional promoter Gal10/Gal1 using the backbone of a pESC-Leu2d vector (Fig 2).

The candidates GmC4H2, GmC4H14 and GmC4H20 were cloned into pESC-Leu2d-LjCPR1-GW destination vector and yeast microsomes overexpressing GmC4Hs were used to check their ability to convert *trans*-cinnamic acid to *p*-coumaric acid. All the parameters for the microsome preparation and the enzyme assays were kept constant for the three candidates. Our results revealed that all GmC4Hs are able to convert *trans*-cinnamic acid to *p*-coumaric acid but with varying efficiencies. Among the three GmC4Hs, GmC4H14 showed highest efficiency to catalyze the reaction followed by GmC4H2 (Fig 7). Yeast microsomes expressing empty vector was used as a negative control where no product (*p*-coumaric acid) formation was observed. The apparent Km value of GmC4H2, GmC4H14 and GmC4H20 for *trans*-cinnamic acid observed were 6.438 ± 0.74, 2.74 ± 0.18 and 3.83 ± 0.44 μM, respectively, with Vmax values of 3.6 ± 0.15, 56.38 ± 0.73 and 0.13 nmoles/min/mg total protein, respectively. Among the three GmC4Hs, GmC4H14 showed highest catalytic activity with Vmax/Km value of 51.44±1.36, while GmC4H2 and GmC4H2 displayed Vmax/Km value of 1.4 ± 0.7 and 0.09, respectively. Our results demonstrated that soybean contains three functional C4Hs and that the class I GmC4Hs are more efficient enzymes for the conversion of *trans*-cinnamic acid to *p*-coumaric acid compared to class II GmC4H.

## Discussion

C4H is one of the key enzymes in the general phenylpropanoid pathway that provides precursor molecule for the synthesis of a myriads of specialized metabolites such as isoflavonoid glyceollins, flavonoids, lignins, proanthocyanins and anthocyanins [48] that play important roles in plant growth and development, and adaptation to environmental perturbations. In this study, we report the functional characterization of *GmC4H* by performing their genome-wide identification, investigation of gene expression, protein in subcellular compartment and comparison of catalytic activities of its isoforms. Our results demonstrate that *GmC4H* gene family members display tissue-specific gene expression and differ in their catalytic activity.

*C4Hs* generally exist as a small gene family. For example, *A. thaliana* [49], *Scutellaria baicalensis* [50] and *Petroselinum crispum* [51] each contain a single *C4H* in their genome while two, three and four C4H encoding genes have been reported in *Brassica napus* [52], *Populus kitakamiensis* [53] and *Populus trichocarpa* × *Populus deltoides* [8], respectively. Soybean contains 346 GmP450s among which three belong to CYP73 family [33]. Since CYP73 family members are C4H in other plants, we suspected three *GmC4H* genes in soybean genome. However, our genome-wide search identified four *GmC4H*s (Table 1, Fig 3) that are categorized into two classes, class I and class II [49, 54, 55]. Previously, a comparative study of C4Hs in soybean, common bean and Arabidopsis reported a total of four *C4H* genes in soybean [48]. However, we observed discrepancies in the locus ID in our study (Table 1) and previous report, which could have resulted from the recent re-annotation of the soybean genome.

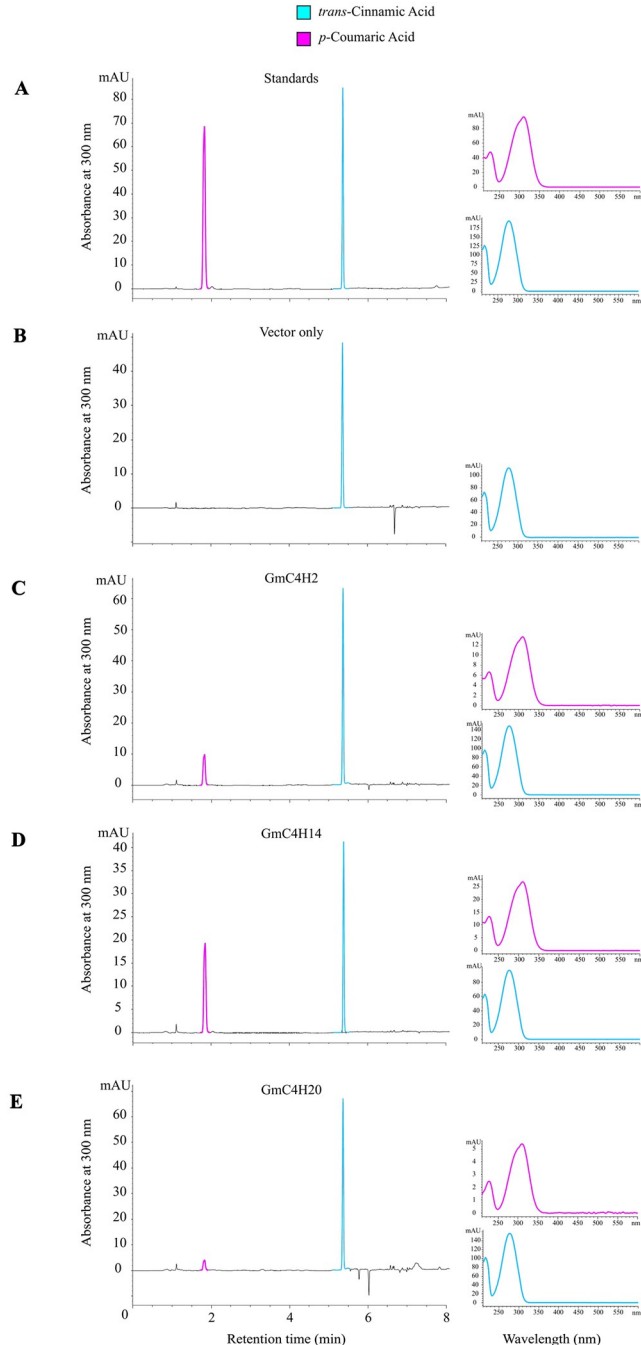

**Fig 7. Functional characterization of candidate GmC4Hs *in vitro*.** Microsomal fraction containing candidate GmC4Hs and LjCPR1 was incubated with *trans*-cinnamic acid and NADPH followed by ethyl acetate extraction. The extract was analyzed by high-performance liquid chromatography (HPLC). (A) standards. (B-E) microsomal proteins + substrate. Spectrum for the substrate and the products are shown on the right.

Based on the phylogenetic analysis, GmC4H2 and GmC4H14 grouped with class I whereas GmC4H10 and GmC4H20 clustered with class II CYP73 family proteins (Fig 4). The diversification of these two classes of CYP73 has been expected to occur through gene duplication early in the evolution of vascular plants. Even though multiple functionally characterized class

**Table 1. Characteristics of soybean *C4H* genes family.**

| Gene Name | Locus name | Gene Location | Splice variants | Coding sequence (nt) | Protein molecular mass (kDa) | Protein isoelectric point (pI) | Predicted subcellular Localization |
|---|---|---|---|---|---|---|---|
| *GmC4H2* | Glyma.02G236500 | Gm02:44245634..44249943 | 1 | 1518 | 58.01 | 8.835 | Endoplasmic reticulum |
| *GmC4H14* | Glyma.14G205200 | Gm14:47892073..47896155 | 1 | 1515 | 57.97 | 8.567 | Endoplasmic reticulum |
| *GmC4H20* | Glyma.20G114200 | Gm20:35555563..35557874 | 1 | 1617 | 61.31 | 8.424 | Endoplasmic reticulum |
| *GmC4H10* | Glyma.10G275550 | Gm10:49904744..49905768 | 1 | 600 | 22.33 | 5.19 | Cytoplasm |

I C4Hs have been found, it contains only two legume C4Hs, *M. sativa* [24] and *C. arietinum* [7] that share 87% sequence identity at the amino acid level with GmC4H2 and GmC4H14. Furthermore, only a single class II CYP73 has been characterized so far [*Camellia sinensis* (Uniprot: ASU87408)] [45]. Several angiosperms such as Arabidopsis, *Helianthus annuus*, *Camptotheca acuminata*, *Pyrus bretschneideri* and plants belonging to Brassicaeae do not contain class II CYP73 [5] suggesting this class of CYP73 play more specialized function and are not required for normal plant growth and development.

Analysis of *GmC4H* gene expression revealed that *GmC4H2*, *GmC4H14* and *GmC4H20* transcripts accumulated at the highest level in root tissue. While a varying degree of tissue and developmental stage-specific gene expression was observed for the members of *GmC4H* family, *GmC4H10* transcript was not detected in any of the tissues used in this study. Our results suggest that *GmC4H10* possibly does not produce a functional protein as the predicted protein sequence lacks PPGP motif and contains a Arg to Lys substitution in the PERF domain (Fig 3). Both these motifs are critical for the efficacy of a P450 [56–58]. Furthermore, *trans*-cinnamic acid substrate docking with GmC4H10 was away from the heme group (Fig 5) providing an additional evidence that it may not be functional.

Previously we reported that the expression of the downstream genes in phenylpropanoid biosynthesis (*CHS7*, *CHS8* and *IFS2*) were found higher at 70 DAP embryo and their expression profile correlated with isoflavonoid accumulation in soybean seeds [59]. The expression of two class I *GmC4H*s, *GmC4H2* and *GmC4H14*, increased dramatically during late embryo developmental stages possibly demonstrating how plants divert their energy in regulating and storing specialized metabolites in the sink tissue. In contrast, *GmC4H20* showed a gradual increase in the level of expression followed by a decline during the late seed developmental stages, thus, suggesting a distinct role of C4Hs belonging to the two different classes. Further studies are required to explore the functions of C4H members belonging to different classes.

For the functional characterization of legume P450s including the GmC4H with ease, we modified a yeast expression vector that contained a CPR from *L. japonicus* along with a gateway cassette (Fig 2). Yeast microsomes expressing GmC4Hs, GmC4H2 and GmC4H14 in the above-engineered custom vector were utilized for the enzyme assay. As shown in Fig 7, all three GmC4Hs are able to catalyze the reaction and convert *trans*-cinnamic acid to *p*-coumaric acid albeit with different efficiency. The observed differences among GmC4H isoforms in their enzymatic reaction were due to the discrepancies in protein expression in yeast or their catalytic efficacy is not yet clear. Despite the weak enzymatic activity of GmC4H20, it is the first reported class II C4H from legume. Class II C4Hs have been suggested to function in lignification process in a number of plant species [60, 61].

In conclusion, we identified three new C4Hs in the soybean genome and functionally characterized them for their activity to convert *trans*-cinnamic acid to *p*-coumaric acid. The amino acid sequences of GmC4H2 and GmC4H14 shared higher sequence identity with bona fide class I C4Hs from legumes and other plant species. The GmC4H20 with class II C4Hs from

other plant species provided an insight into their close evolutionary relationship with other C4Hs from land plants. We also engineered an easy-to-clone gateway compatible yeast expression vector for functional characterization of P450s.

## Supporting information

**S1 Fig. Protein-ligand interaction at the GmC4H active site of *trans*-cinnamic acid.**
(TIF)

**S1 Table. List of primer used for vector construction, gene cloning and qPCR.**
(DOCX)

**S2 Table. Details of predicted and characterized CYP73 members used in phylogenetic analysis.**
(XLSX)

## Acknowledgments

The authors thank Dr. Dae-Kyun Ro (University of Calgary) for pESC-Leu2d clone, Dr. Krzysztof Szczyglowski for *L. japonicus* RNA, and Jordan Vanderburt, Tim McDowell and Kuflom Kuflu for technical assistance.

## Author Contributions

**Conceptualization:** Sangeeta Dhaubhadel.

**Data curation:** Praveen Khatri.

**Formal analysis:** Praveen Khatri, Ling Chen, Sangeeta Dhaubhadel.

**Funding acquisition:** Sangeeta Dhaubhadel.

**Investigation:** Ling Chen, Sangeeta Dhaubhadel.

**Methodology:** Praveen Khatri.

**Project administration:** Praveen Khatri, Sangeeta Dhaubhadel.

**Resources:** Istvan Rajcan.

**Supervision:** Sangeeta Dhaubhadel.

**Writing – original draft:** Praveen Khatri.

**Writing – review & editing:** Istvan Rajcan, Sangeeta Dhaubhadel.

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
