## [Decision Letter · Decision Letter 0]

12 Apr 2023

PONE-D-23-07065Functional characterization of Cinnamate 4-hydroxylase gene family in soybean (Glycine max)PLOS ONE

Dear Dr. Dhaubhadel,

Thank you for submitting your manuscript to PLOS ONE. After careful consideration, we feel that it has merit but does not fully meet PLOS ONE’s publication criteria as it currently stands. Therefore, we invite you to submit a revised version of the manuscript that addresses the points raised during the review process. Please submit your revised manuscript by May 27 2023 11:59PM. If you will need more time than this to complete your revisions, please reply to this message or contact the journal office at plosone@plos.org. Please include the following items when submitting your revised manuscript:A rebuttal letter that responds to each point raised by the academic editor and reviewer(s). You should upload this letter as a separate file labeled 'Response to Reviewers'.A marked-up copy of your manuscript that highlights changes made to the original version. You should upload this as a separate file labeled 'Revised Manuscript with Track Changes'.An unmarked version of your revised paper without tracked changes. You should upload this as a separate file labeled 'Manuscript'.If applicable, we recommend that you deposit your laboratory protocols in protocols.io to enhance the reproducibility of your results. Protocols.io assigns your protocol its own identifier (DOI) so that it can be cited independently in the future. For instructions see: https://journals.plos.org/plosone/s/submission-guidelines#loc-laboratory-protocols. Additionally, PLOS ONE offers an option for publishing peer-reviewed Lab Protocol articles, which describe protocols hosted on protocols.io. Read more information on sharing protocols at https://plos.org/protocols?utm_medium=editorial-email&utm_source=authorletters&utm_campaign=protocols.

We look forward to receiving your revised manuscript.

Kind regards,

Mojtaba Kordrostami, Ph.D.

Academic Editor

PLOS ONE

Journal Requirements:

"The authors thank Dr. Dae-Kyun Ro (University of Calgary) for pESC-Leu2d clone, Dr. Krzysztof Szczyglowski for L. japonicus RNA, and Jordan Vanderburt, Tim McDowell and Kuflom Kuflu for technical assistance. This research was supported by the Agriculture and Agri-Food Canada’s Genomics Research and Development Initiative grant (J-002364) and the ASC-09 Soybean Cluster Activity #7A (J-002060) to SD"

"This research was supported by the Agriculture and Agri-Food Canada’s Genomics Research and Development Initiative grant (J-002364) and the ASC-09 Soybean Cluster Activity #7A (J-002060) to SD."

Additional Editor Comments (if provided):

Dear collegues

I have received the reports from 2 experts in this field. As you can see it needs revisions before final acceptance.

Please answer the comments and sent back the revised version. Please also recheck the English by a native person.

Regards

Reviewers' comments:

Reviewer's Responses to Questions

**Comments to the Author**

1. Is the manuscript technically sound, and do the data support the conclusions?

Reviewer #1: Yes

Reviewer #2: Yes

2. Has the statistical analysis been performed appropriately and rigorously? 

Reviewer #1: Yes

Reviewer #2: N/A

3. Have the authors made all data underlying the findings in their manuscript fully available?

Reviewer #1: Yes

Reviewer #2: Yes

4. Is the manuscript presented in an intelligible fashion and written in standard English?

Reviewer #1: Yes

Reviewer #2: Yes

5. Review Comments to the Author

Reviewer #1: “Functional characterization of Cinnamate 4-hydroxylase gene family in soybean (Glycine max)” is a massive piece of work and it has provided good information. I thank the authors for this work. The introduction provides comprehensive informations on the background. The research was properly planned and performed. Methods and results are well described. I recommend this manuscript for publication.

Reviewer #2: Khatri et. al characterized the soybean C4H genes and confirmed cinnamate hydroxylation activity by enzymatic assay with heterologously expressed protein. The authors also provided a cloning strategy for the coexpression of a plant CPR with P450 for future functional study. The authors claimed they have identified the C4H genes across the manuscript, which have already been reported previously. They should take less credits on that or just claim the locus ID following reannotated of soybean genome was updated.

Line 275&line 365 The docking was done for GmC4H10 and the authors reported docking energy. Thus, instead of saying “not observed”, it is better to claim the docking position is impossible for hydroxylation.

Line 308 Specifically, only class II P450 requires CPR.

Line 322 It could be fairer to compare the catalytic efficiency (Vmax/Km). The full kinetic assay (Michaelis–Menten fitting) with three C4Hs is more informative than figure 7, and figure 7 can be moved to supplementary figure.

Line 392 Has the catalytic efficiency been compared to that with soybean CPR?

Line 387 There is an underlined o in coumaric acid.

6. PLOS authors have the option to publish the peer review history of their article (what does this mean?). If published, this will include your full peer review and any attached files.

Reviewer #1: No

Reviewer #2: No

---

## [Author Response · Author response to Decision Letter 0]

25 Apr 2023

Response to reviewers Comments

Reviewer #1: “Functional characterization of Cinnamate 4-hydroxylase gene family in soybean (Glycine max)” is a massive piece of work and it has provided good information. I thank the authors for this work. The introduction provides comprehensive informations on the background. The research was properly planned and performed. Methods and results are well described. I recommend this manuscript for publication.

Response: Thank you

Reviewer #2: Khatri et. al characterized the soybean C4H genes and confirmed cinnamate hydroxylation activity by enzymatic assay with heterologously expressed protein. The authors also provided a cloning strategy for the coexpression of a plant CPR with P450 for future functional study. The authors claimed they have identified the C4H genes across the manuscript, which have already been reported previously. They should take less credits on that or just claim the locus ID following reannotated of soybean genome was updated.

Response: Thank you for your valuable feedback. The main objective of our study was to characterize the function of C4Hs in soybean and develop a vector that can aid in the characterization of P450s, specifically in legumes. While previous study by Y. Reinprecht et al. (2017) identified putative C4H candidates, our study aimed to provide functional validation of C4Hs in soybean using enzyme assays, as well as to study the dynamics of soybean C4Hs.

Line 275&line 365 The docking was done for GmC4H10 and the authors reported docking energy. Thus, instead of saying “not observed”, it is better to claim the docking position is impossible for hydroxylation.

Response: As suggested by the reviewer, we have amended the statement.

Line 308 Specifically, only class II P450 requires CPR.

Response: As suggested by the reviewer, we have revised the sentence to emphasize only eukaryotic P450 (class II) require CPR.

Line 322 It could be fairer to compare the catalytic efficiency (Vmax/Km). The full kinetic assay (Michaelis–Menten fitting) with three C4Hs is more informative than figure 7, and figure 7 can be moved to supplementary figure.

Response: Thank you for your suggestion regarding the comparison of catalytic efficiency for the three C4Hs in our study. We agree that comparing Vmax/Km is a more comprehensive way of comparing the catalytic activity of the enzymes. Therefore, we have performed the full kinetic assay (Michaelis-Menten fitting) as suggested and the results are consistent with our initial findings. We would like to keep figure 7 in the main paper as it is a visual representation of our finding that one of the three C4Hs is more efficient in soybean, and we believe it adds value to the paper. We hope this will be acceptable. 

Line 392 Has the catalytic efficiency been compared to that with soybean CPR?

Response: We are unsure of the reviewer’s query here. As per our understanding, it was not possible for us to compare the efficiency with soybean CPRs because we used the product formation measurement method in the enzyme assay instead of the NADPH oxidation method, which can be used to analyze the kinetics of CPRs. In our study, we used Lotus japonicus CPR1 for all three C4Hs, and therefore, we believe that the kinetic parameters are comparable. LjCPR1 has been used in several studies as below:

Seki, H., Sawai, S., Ohyama, K., Mizutani, M., Ohnishi, T., Sudo, H., Fukushima, E.O., Akashi, T., Aoki, T., Saito, K., and Muranaka, T. (2011). Triterpene Functional Genomics in Licorice for Identification of CYP72A154 Involved in the Biosynthesis of Glycyrrhizin. The Plant Cell 23, 4112-4123.

Yano, R., Takagi, K., Takada, Y., Mukaiyama, K., Tsukamoto, C., Sayama, T., Kaga, A., Anai, T., Sawai, S., Ohyama, K., Saito, K., and Ishimoto, M. (2017). Metabolic switching of astringent and beneficial triterpenoid saponins in soybean is achieved by a loss-of-function mutation in cytochrome P450 72A69. The Plant Journal 89, 527-539.

Line 387 There is an underlined o in coumaric acid.

Response: Corrected.

---

## [Editor Report · Decision Letter 1]

28 Apr 2023

Functional characterization of Cinnamate 4-hydroxylase gene family in soybean (Glycine max)

PONE-D-23-07065R1

Dear Dr. Dhaubhadel,

We’re pleased to inform you that your manuscript has been judged scientifically suitable for publication and will be formally accepted for publication once it meets all outstanding technical requirements.

Kind regards,

Mojtaba Kordrostami, Ph.D.

Academic Editor

PLOS ONE

Additional Editor Comments (optional):

Dear authors,

I am pleased to inform you that your manuscript titled "Functional characterization of Cinnamate 4-hydroxylase gene family in soybean (Glycine max)" has been accepted for publication in PLOS One.

Your study provides valuable insights into the functional characterization of the Cinnamate 4-hydroxylase gene family in soybean, which will undoubtedly contribute to our understanding of soybean genetics and ultimately benefit the agricultural industry.

We appreciate your contribution to our journal and look forward to publishing your article.

Thank you for choosing PLOS One as the platform for sharing your research.

Best regards,
---

## [Editor Report · Acceptance letter]

5 May 2023

PONE-D-23-07065R1 

Functional characterization of *Cinnamate 4-hydroxylase* gene family in soybean (*Glycine max*) 

Dear Dr. Dhaubhadel:

I'm pleased to inform you that your manuscript has been deemed suitable for publication in PLOS ONE. Congratulations! Your manuscript is now with our production department. 

Kind regards, 

on behalf of

Dr. Mojtaba Kordrostami 

Academic Editor

PLOS ONE